# Neurosurgical Defeats: Critically Ill Patients and the Role of Palliative Care Service

**DOI:** 10.3390/jpm12101565

**Published:** 2022-09-23

**Authors:** Quintino Giorgio D’Alessandris, Maria Adelaide Ricciotti, Davide Palombi, Ludovico Agostini, Pier Paolo Mattogno, Giuseppe Maria Della Pepa, Alessio Albanese, Silvia Chiesa, Sabrina Dispenza, Eleonora Meloni, Anita Maria Tummolo, Roberto Pallini, Christian Barillaro, Alessandro Olivi, Liverana Lauretti

**Affiliations:** 1Department of Neurosurgery, Fondazione Policlinico Universitario A. Gemelli IRCCS, Università Cattolica del Sacro Cuore, Largo A. Gemelli, 8, 00168 Rome, Italy; 2Department of Palliative Care and Continuity of Care, Fondazione Policlinico Universitario A. Gemelli IRCCS, Università Cattolica del Sacro Cuore, Largo A. Gemelli, 8, 00168 Rome, Italy; 3Department of Radiation Oncology, Fondazione Policlinico Universitario A. Gemelli IRCCS, Università Cattolica del Sacro Cuore, Largo A. Gemelli, 8, 00168 Rome, Italy

**Keywords:** brain tumor, performance status, multidisciplinary tumor board, neurosurgery, palliative care

## Abstract

The onco-functional balance in neuro-oncology requires maximizing tumor removal while rigorously preserving patients’ neurological status. When postoperative worsening prevents the implementation of oncologic treatments, palliative care service offers an individualized path for symptom and psychosocial distress relief. Here, we report on a series of 25 patients operated on for malignant brain tumor who did not undergo adjuvant treatments after neurosurgery; they represented 3.9% of the whole institutional surgical series. These patients were significantly older and had a lower preoperative Karnofsky performance status than the whole cohort. Importantly, in 22 out of 25 (88%) cases, a surgical complication occurred, leading to clinical worsening in 21 patients. For the end of life, the majority of patients chose a hospice care facility (72%). While a careful selection of brain tumor patients candidate to neurosurgery is required, palliative care service provided invaluable help in coping with patients’ and caregivers’ needs.

## 1. Introduction

Primary cerebral malignancies account for 1% of all invasive cancer cases in the United States. In this country, about 24,530 new diagnoses of primary brain malignancies and 18,600 deaths were registered in 2021 [1]. The 5th edition of the WHO classification of central nervous system tumors, issued in 2021 [2], introduced profound changes and has reinforced the role of molecular profile of tumors in defining the correct “integrated” diagnosis. The most frequent and most aggressive brain tumor in adults is glioblastoma, IDH-wildtype, whose incidence is 5 cases per 100,000 persons/year and whose median overall survival, despite multimodal aggressive treatment, does not exceed 15–18 months [2,3]. Metastatic brain malignancies are more frequent than primary brain cancers; they originate mainly from lung cancer, breast cancer, and melanoma, and their incidence has increased in recent years, in parallel with the improved prognosis of primary diseases [2].

Brain tumors deeply affect the quality of life of patients [4], and antitumor treatments, particularly surgery, add general and neurological risks to an already frail patient. The onco-functional balance, consisting of maximizing tumor removal with the preservation of neurological functions, is a fundamental concept in neurosurgical oncology [3]. The indication for neurosurgery for patients whose neurological status is already compromised is debated since it is known that non-functionally independent patients face huge difficulties in undergoing adjuvant treatments [3]. Therefore, the therapeutic strategy for patients with a low performance status is generally discussed in multidisciplinary meetings in which the option of palliative care is evaluated. In summary, neurosurgeons tend to operate on brain tumor patients who are preoperatively fit to bear the burden of oncological treatments and who are presumed to keep their good conditions postoperatively. Unfortunately, notwithstanding this selection process, after neurosurgery some patients cannot undergo adjuvant treatment [5]. For these cases, palliative care has a fundamental role to relieve the patients’ and caregivers’ burdens and to define the most appropriate clinical path for each patient.

The goal of the present work is to describe our series of patients that were sent to palliation after neurosurgery, with the aim to identify the key features of this group and to underline the priceless role of the palliative care service.

## 2. Materials and Methods

### 2.1. Patient Enrollment

We retrospectively reviewed the charts of patients surgically treated at the Department of Neurosurgery, Fondazione Policlinico Universitario A. Gemelli IRCCS, Rome, Italy, for malignant brain tumors in the timeframe of 2018–2021. Both primary and metastatic tumors were screened. Patients were enrolled in the present study if they did not undergo adjuvant therapy after neurosurgery and only palliative care was indicated. However, oral temozolomide given only for palliative purposes was not considered an exclusion criterion. All these patients were chronically treated with dexamethasone (4–16 mg/day). Demographic, clinical, surgical, and follow-up data of these patients were recorded.

### 2.2. Statistical Analysis

Data are presented for continuous variables as means ± SD or median (range) and for categorical variables as absolute and relative frequencies. Paired and unpaired *t*-tests were computed to compare the continuous variables between groups. StatView ver. 5.0 (SAS Institute, Cary, NC, USA) was used for the analyses.

## 3. Results

In the study period, 639 patients underwent a craniotomy for a malignant brain tumor (either primary or metastatic) at our institution. The mean patient age was 61.2 ± 11.7 years; the median preoperative and postoperative Karnofsky Performance Status (KPS) values were both 70. Among these, 25 patients (3.9%) did not undergo adjuvant treatments after surgery and were enrolled in the present study (Figure 1). The demographics of these patients are presented in Table 1. The mean age was 67.3 ± 9.3 years. Thus, the patients in this group were significantly older than the whole cohort (*p* = 0.017, unpaired *t*-test). Seventeen patients (68%) harbored a glioblastoma, IDH-wildtype, and eight patients (32%) harbored a brain metastasis (a single tumor in two cases and multiple tumors in six cases). Deep-seated tumors accounted for 40% of all cases. The median preoperative KPS was 60, and nine patients (36%) had a KPS ≤ 40. The preoperative KPS was thus significantly lower than that in the whole cohort of 639 patients (*p* < 0.0001, unpaired *t*-test). Three of these cases harbored hemorrhagic tumors and were operated in an emergency setting. The other patients harbored large tumors for which a postoperative improvement of clinical conditions could be expected; in these cases, the surgical indication was established after extensive multidisciplinary discussions involving patients’ caregivers.

Table 2 shows the clinical and surgical features of the patients. In five patients harboring deep or multicentric glioma, only neurosurgical biopsies (four stereotactic frameless needle biopsies and one endoscopic endoventricular biopsy) were performed. The other patients underwent craniotomy for resective purposes. Ultrasound aspiration was used in all cases; intraoperative fluorescence (5-ALA) was used in 5/12 (41.7%) GBM cases. The median postoperative KPS was 30. Thus, it was significantly reduced compared to the preoperative value (*p* < 0.0001, paired *t*-test). Obviously, the postoperative KPS was also lower compared to the whole cohort (*p* < 0.0001, unpaired *t*-test). A surgical complication occurred in 22 cases (88%), leading to a worsening of KPS in 21 cases (84%). The most frequent complication was stroke (44%), mainly due to injury to perforating arteries.

The patient follow-up data are shown in Table 3. We had five patients who died during hospitalization within the first month after surgery. In all cases, the in-hospital palliative care service took care of the patient’s course. This involved a meeting with the patient’s relatives and caregivers in order to analyze their needs and to find the best solution to cope with them. Further meetings were scheduled to meet the psychological needs of the patients and their relatives. At the conclusion of this process, only two patients had homecare, while the majority of them (72%) went to a hospice for end-of-life care. The median overall survival after surgery was 52 days.

## 4. Discussion

In this work, we aimed to reliably describe a cohort of brain tumor patients who were not able to undergo adjuvant treatments after surgery because of clinical worsening. These patients made up a small percentage of most surgical series and were less than 4% (3.9%) in ours; however, such cases deserve to be thoroughly dissected to achieve an effective personalized plan of care. In the paper, we deeply analyzed the following aspects:

*Preoperative KPS and age.* The patients in our study group were older (69.3 vs. 61.2 mean age) and had a lower preoperative KPS (60 vs. 70) than the entire group of malignant brain tumor patients that underwent neurosurgery in the same timeframe. This raises the issue of correct patient selection for neurosurgery. However, older and unfit patients represented a non-negligible portion of the whole cohort of 639 brain tumor patients; about 15% were 75 years or older, and about 13% had a KPS of 60 or lower. Still, the vast majority of those patients were able to complete the standard adjuvant treatment. This observation relativizes the importance of plain numbers; though the KPS score and age are important prognostic parameters, indication for surgery in fragile patients should be decided by the multidisciplinary brain tumor board, as routinely happens at our institution [6]. During the multidisciplinary meeting, the patient’s clinical history is thoroughly illustrated by the referral physician, and all possible therapeutic options are evaluated. Such a selection assures that surgical failures are reduced to the largest possible extent.

*Perioperative complications.* Another key point from our analysis is the detrimental role of surgical complications that reduced the KPS in more than 80% of patients of our study group. Among them, ischemic stroke was the most frequent complication (44%), mostly due to vascular injury to deep perforating arterial branches. The wide use of intraoperative tools to obtain a gross total tumor removal (5-ALA, ultrasound aspiration, etc.) might augment the risk of forcing resection beyond safety; indeed, neurophysiological monitoring is an indispensable useful warning for the surgeon dealing with tumors in eloquent areas. Moreover, it is widely known that a low postoperative KPS is one of the strongest negative predictors of unsatisfactory outcomes in brain tumors [7], and current guidelines suggest that the functional outcome should have priority over the oncologic outcome, particularly in malignant tumors [3].

*Management of end of life.* Palliative care has recently been introduced in the setting of brain tumors [8], with the goal to improve patients’ and caregivers’ quality of life. The early implementation of palliative care is requested by the reduced cognitive ability of brain tumor patients [9] and by the heavy distress that is experienced by caregivers. Moreover, in the context of the heavy socioeconomic burden of cancer [10,11], it has been shown that early palliative care activation reduces the costs of hospitalizations [12] and leads to an increase in home discharge for end-of-life care [13]. In most of the cases discussed here, caregivers had to face a sudden worsening of patients’ neurological conditions without having the possibility to set up adequate domestic assistance: this can explain the low rate of home deaths in our series. On the other hand, palliative care service consultation was of tremendous help in allowing patients’ relatives to accept the irreversible clinical deterioration of their loved ones. In fact, few patients died in the acute ward, while the majority of them accepted a traditional palliative care path in a hospice.

The limitations of the present study include its monoinstitutional design, the relatively small number of patients, and the lack of detailed psychological, cognitive, and social assessments of patients and their caregivers. However, we think that our study could be of help for neurosurgeons and neuro-oncologists aiming at personalizing patient treatment, and our study adds evidence to the growing literature on palliation in brain tumors.

## Figures and Tables

**Figure 1 jpm-12-01565-f001:**
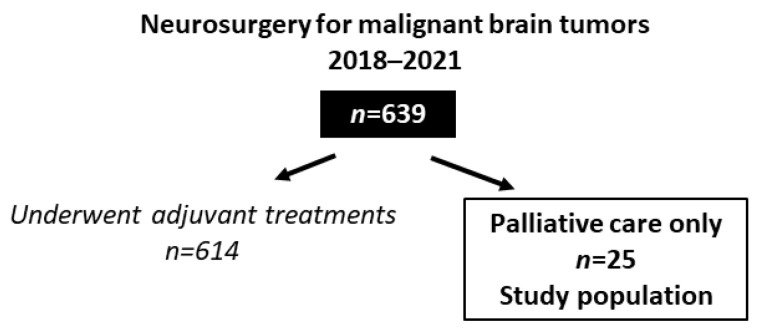
Study flowchart.

**Table 1 jpm-12-01565-t001:** Demographics of enrolled patients.

Parameter	Value
*n*	25
Age, mean	67.3 ± 9.3
Sex, M:F (%)	72%:28%
Histology	
Glioblastoma (%)	17 (68%)
Metastasis	8 (32%)
Lung (adenocarcinoma)	5 (62.5%)
Lung (squamous carcinoma)	1 (12.5%)
Kidney	1 (12.5%)
Colorectal	1 (12.5%)
Deep-seated tumors	10 (40%)
Multicentric/multiple tumors	10 (40%)
Preoperative KPS, median (range)	60 (30–70)

**Table 2 jpm-12-01565-t002:** Clinical and surgical features.

Parameter	Value
Surgery	
Resection	20 (80%)
Biopsy	5 (20%)
Stereotactic frameless	4 (80%)
Endoventricular endoscopic	1 (20%)
First Diagnosis	20 (80%)
Recurrent Tumor	5 (20%)
Postoperative KPS, median (range)	30 (0–40)
Complications	
Hemorrhage	6 (24%)
Stroke	11 (44%)
Infection	7 (28%)
Hydrocephalus	4 (16%)
Pulmonary Embolism	1 (4%)

**Table 3 jpm-12-01565-t003:** Outcomes.

Parameter	Value
End-of-life setting, n (%)	
Neurosurgery ward	5 (20%)
Hospice	18 (72%)
Home	2 (8%)
Palliative TMZ, *n* (%)	1 (4%)
In-hospital palliative care service	25 (100%)
Overall survival, median (range) (days)	52 (6–205)

## Data Availability

The source data are available from the corresponding author upon reasonable request.

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
