# Peer review of "Neurosurgical Defeats: Critically Ill Patients and the Role of Palliative Care Service"

_jpm, 2022, doi:10.3390/jpm12101565_

Round 1
Reviewer 1 Report
The original article by D’Alessandris et al. "Neurosurgical defeats: critically ill patients and the role of palliative care service" covers a potentially interesting and emerging topic related to the management of patients with brain tumors. In this sense, this remains to be potentially interesting for the Journal of Personalized Medicine readers. I regard the main point of this paper as highly attractive as well as the results are clearly presented. The
text does not contain any major errors, therefore I have some minor comments and recommendations:
1. There is a need to provide Figure/Chart/Diagram summarizing patients recruitment
2. More expanded introduction regarding the epidemiology/pathogenesis data regarding the brain tumors is needed
3. Demographics of enrolled patients should included type of brain metastasis (origin),
4. Clinical and surgical features should contain type of biopsy performed (open, stereotactic, navigation)
5. 5-ALA, ultrasound apiration another tools should be mentioned in the statistic
6. Following references should be added and properly cited within the main text:
- Mela A, Poniatowski ŁA, Drop B, Furtak-Niczyporuk M, Jaroszyński J, Wrona W, Staniszewska A, Dąbrowski J, Czajka A, Jagielska B, Wojciechowska M, Niewada M. Overview and Analysis of the Cost of Drug Programs in Poland: Public Payer Expenditures and Coverage of Cancer and Non-Neoplastic Diseases Related Drug Therapies from 2015-2018 Years. Front Pharmacol. 2020 Aug 14;11:1123. doi: 10.3389/fphar.2020.01123.
- Lovely MP, Stewart-Amidei C, Page M, Mogensen K, Arzbaecher J, Lupica K, Maher ME. A new reality: long-term survivorship with a malignant brain tumor. Oncol Nurs Forum. 2013 May 1;40(3):267-74. doi: 10.1188/13.ONF.267-274.
- Jaroszynski J, Mela A, Furtak-Niczyporuk M, et al. Off-Label Use OfMedicinal Products - Legal Rules And Practices.Acta Pol Pharm.2019;76(4):621-628. https://doi.org/10.32383/appdr/105795
- Brain D, Jadambaa A. Economic Evaluation of Long-Term Survivorship Care for Cancer Patients in OECD Countries: A Systematic Review for Decision-Makers. Int J Environ Res Public Health. 2021 Nov 3;18(21):11558. doi: 10.3390/ijerph182111558.
7. In some places the use of English could be improved on.
Completing this gaps will have an impact on the understanding the aim of the study and, from my point of view, is absolutely necessary.
Reviewer 2 Report
The manuscript entitled "Neurosurgical defeats: critically ill patients and the role of palliative care service" by Quintino Giorgio D'Alessandris and collaborators contains a retrospective analysis of a series of 25 brain tumors (both glioblastomas and metastasis) treated by Neurosurgeons between 2018 and 2021 in the authors' Institution. Those patients that amounted to 3.9% of all brain tumor patients operated in those years, did not undergo adjuvant therapy after neurosurgery and were sent directly to palliative care services.
The study is purely retrospective with minimal follow-up since the last subjects were included in 2021 and the patient series is relatively small and highly heterogeneous with little power to reach strong clinical conclusions except the illustration of the practice of their Institution.
I believe that a longer period of accrual and the addition of data relative to patient survival after Neurosurgery will significantly improve the impact of the paper. For metastatic tumors please add the pathological diagnosis and the oncological grading of the tumors before neurosurgery.
Some indications about the use and the dose of steroids will also be useful.
Minor points
page 4 row 130 "...numbers: KPS score and age are important prognostic 117 indicators but surgery for complex patients, as we routinely do in our institution" please clarify.
